# Therapeutic Anti-KIR Antibody of 1–7F9 Attenuates the Antitumor Effects of Expanded and Activated Human Primary Natural Killer Cells on In Vitro Glioblastoma-like Cells and Orthotopic Tumors Derived Therefrom

**DOI:** 10.3390/ijms241814183

**Published:** 2023-09-16

**Authors:** Ryosuke Maeoka, Tsutomu Nakazawa, Ryosuke Matsuda, Takayuki Morimoto, Yoichi Shida, Shuichi Yamada, Fumihiko Nishimura, Mitsutoshi Nakamura, Ichiro Nakagawa, Young-Soo Park, Takahiro Tsujimura, Hiroyuki Nakase

**Affiliations:** 1Department of Neurosurgery, Nara Medical University, Nara 634-8521, Japan; r.maeoka@naramed-u.ac.jp (R.M.); t.morimoto@naramed-u.ac.jp (T.M.); yoichi_0723@yahoo.co.jp (Y.S.); syamada@naramed-u.ac.jp (S.Y.); fnishi@naramed-u.ac.jp (F.N.); mnaka@grandsoul.co.jp (M.N.); nakagawa@naramed-u.ac.jp (I.N.); park-y-s@naramed-u.ac.jp (Y.-S.P.); nakasehi@naramed-u.ac.jp (H.N.); 2Grandsoul Research Institute for Immunology, Inc., Uda 633-2221, Japan; takahiro@grandsoul.co.jp; 3Clinic Grandsoul Nara, Uda 633-2221, Japan

**Keywords:** NK cell, glioblastoma, immunotherapy, KIR, GiNKs

## Abstract

Glioblastoma (GBM) is the leading malignant intracranial tumor, where prognosis for which has remained extremely poor for two decades. Immunotherapy has recently drawn attention as a cancer treatment, including for GBM. Natural killer (NK) cells are immune cells that attack cancer cells directly and produce antitumor immunity-related cytokines. The adoptive transfer of expanded and activated NK cells is expected to be a promising GBM immunotherapy. We previously established an efficient expansion method that produced highly purified, activated primary human NK cells, which we designated genuine induced NK cells (GiNKs). The GiNKs demonstrated antitumor effects in vitro and in vivo, which were less affected by blockade of the inhibitory checkpoint receptor programmed death 1 (PD-1). In the present study, we assessed the antitumor effects of GiNKs, both alone and combined with an antibody targeting killer Ig-like receptor 2DLs (KIR2DL1 and DL2/3, both inhibitory checkpoint receptors of NK cells) in vitro and in vivo with U87MG GBM-like cells and the T98G GBM cell line. Impedance-based real-time cell growth assays and apoptosis detection assays revealed that the GiNKs exhibited growth inhibitory effects on U87MG and T98G cells by inducing apoptosis. KIR2DL1 blockade attenuated the growth inhibition of the cell lines in vitro. The intracranial administration of GiNKs prolonged the overall survival of the U87MG-derived orthotopic xenograft brain tumor model. The KIR2DL1 blockade did not enhance the antitumor effects; rather, it attenuated it in the same manner as in the in vitro experiment. GiNK immunotherapy directly administered to the brain could be a promising immunotherapeutic alternative for patients with GBM. Furthermore, KIR2DL1 blockade appeared to require caution when used concomitantly with GiNKs.

## 1. Introduction

Glioblastoma (GBM) is the most lethal primary malignant brain tumor in adults and is associated with a poor prognosis. Even with current standard treatments such as surgery, radiation therapy, chemotherapy, and tumor treating fields (TTFields), the median overall survival (mOS) is 14.6–20.9 months from randomization and the 5-year survival rate is <15% [1,2,3,4]. Therefore, novel and effective treatment strategies are needed for patients with GBM, for whom immunotherapy could be a potential treatment option. Immunotherapy using immune checkpoint inhibitor (ICI) drugs has revolutionized the treatment of malignant melanoma and lung, bladder, and renal cancers [5,6,7]. However, GBM is resistant to immunotherapy using ICI drugs, including nivolumab, which is a fully human immunoglobulin (Ig) G4 monoclonal antibody targeting the programmed death 1 (PD-1) immune checkpoint receptor. This resistance is due to the poor antigen-presenting properties of the brain rather than the tumor-intrinsic immunosuppressive properties of GBM tumor cells [8,9]. Moreover, GBM is highly resistant to standard treatments due to a combination of tumor heterogeneity, adaptive expansion of resistant cellular subclones, immune surveillance evasion, and the manipulation of signaling pathways involved in tumor progression and the immune response [10].

Based on the above findings, there is great interest in using the properties of natural killer (NK) cells to develop the next cancer immunotherapy [11]. NK cells were discovered in the 1970s and are critical in first-line host defense against infections and tumors by mediating cytotoxic function and producing cytokines without prior sensitization [12,13,14]. NK cells are innate lymphocytes that act alongside other immune cells in the response against various malignant tumors. Previously, we focused on NK cell function against GBM cells and reported on genuine induced NK cells (GiNKs), which are highly purified human NK cells derived from peripheral blood mononuclear cells (PBMCs) using a feeder-free method and that exhibited high NK activity against GBM cells in vitro [15,16]. The expansion method rapidly yielded a large number of highly purified NK cells. Moreover, the GiNKs exerted an antitumor effect against subcutaneous ectopic GBM-like cell-derived xenograft models in vivo [17].

NK cell function is regulated by multiple types of signals transduced by the activating and inhibitory receptors that recognize ligands expressed on potential target cells [13]. NK cells also demonstrate potent cytotoxic activity against tumor cells by promoting apoptosis [14]. Several activating receptors expressed on NK cells recognize the associated ligands on GBM cells [18,19]. Contrastingly, the inhibitory receptor ligands are associated with NK cell-mediated cytotoxicity against tumor cells [20,21,22]. Multiple immune-suppressive receptors, e.g., killer immunoglobulin (Ig)-like receptors (KIR), PD-1, T cell immunoglobulin mucin family member 3 (TIM3), lymphocyte activation gene 3 (LAG3), T cell immunoreceptor with Ig and ITIM domains (TIGIT), TACTILE (CD96), and transforming growth factor-ß (TGF-ß) receptor, are expressed on NK cells to prevent NK cells from fully exerting antitumor effects [13,23,24,25,26,27,28,29,30,31,32,33].

NK cell activation is negatively regulated by negative signal transduction via KIR, an inhibitory receptor expressed on NK cells that recognizes human leukocyte antigen (HLA) class I molecules [20,34]. The human KIR family comprises polymorphic Ig-like molecules expressed on NK cells and small subsets of CD8^+^ and gd+ T cells [34]. KIR2DL1 and KIR2DL2/3 recognizes distinct HLA-C allotypes or HLA class I, specifically, KIR2DL1 binds HLA-Cw2, -4, -5, and -6, while KIR2DL2/3 bind to -Cw1, -3, -7, and -8 [34]. We previously analyzed and determined the expression of activating and inhibitory receptor ligands and NK cells expressing KIR and their corresponding ligand for HLA-ABC [15] and determined that three glioma cell lines were positive for HLA-ABC. Additionally, two GBM cell lines (T98G and LN-18) and a GBM-like cell line (U87MG) strongly expressed HLA class I. Furthermore, the U87MG GBM-like cell line and the T98G human GBM cell line were positive for HLA-Cw5 and -Cw4/Cw7, respectively [35]. In addition, the isotype of recombinant human anti-KIR2DL1 antibody (1–7F9) was determined to be IgG4, which is known to bind CD64 with lower affinity than other IgG isotypes, and therefore is considered appropriate for a blocking, nondepleting therapeutic monoclonal antibody [34]. 1–7F9 represents a humanized antibody with cross reactivity for KIR2DL1 and KIR2DL2/3. We considered 1–7F9 demonstrate as only anti-KIR2DL1 antibody for U87MG and as both anti-KIR2DL1 antibody and anti-KIR2DL2/3 antibody for T98G. Moreover, KIR2DL1 has the strongest inhibitory power, followed by KIR2DL2/3 among the inhibitory KIRs [36]. Therefore, we postulated that blockade antibody of KIR2DL1, a typical inhibitory KIR, could enhance the antitumor effect of GiNKs on GBM.

In the present study, we evaluated the in vitro antitumor effects of GiNKs in combination with anti-KIR2DL1 on the T98G GBM cell line and the U87MG GBM-like cell line. Furthermore, we evaluated the antitumor effects of GiNKs in combination with anti-KIR2DL1 antibodies using intracranial direct infusion in an orthotopic U87MG derived-xenograft model, which was aimed at clinical application.

## 2. Results

### 2.1. KIR2DL1 and KIR2DL2/3 Expression in Gliomas Based on TCGA Data Set

To confirm the KIR2DL1 expression patterns, we obtained the RNA sequencing data of gliomas from the GlioVis data portal and The Cancer Genome Atlas (TCGA) database [37]. KIR2DL1 was expressed in 3.93 ± 0.18% (mean ± SD converted log2) and 3.81 ± 0.17% of non-tumor and GBM tissue, respectively. KIR2DL2 was expressed in 4.23 ± 0.27% and 4.20 ± 0.23% of non-tumor and GBM tissue, respectively. KIR2DL3 was expressed in 4.69 ± 0.29% and 4.35 ± 0.25% of non-tumor and GBM tissue, respectively. KIR2DL1 and KIR2DL2 expression did not demonstrate differences between GBM and non-tumor samples in TCGA database (*p* = 0.08 and 0.69, respectively), while KIR2DL3 expression demonstrated differences (*p* < 0.01). The Kaplan-Meier curves following log-rank testing demonstrated that low KIR2DL1 expression predicted poor OS with significant differences in TCGA database (*p* = 0.04), while KIR2DL2 and KIR2DL3 expression did not predict poor OS in TCGA database (*p* = 0.29, and 0.33, respectively) (Figure 1A).

### 2.2. KIR2DL1 and KIR2DL2/3 Expression on GiNKs

Flow cytometry analysis detected KIR2DL1 and KIR2DL2/3 expression on the GiNK cell surface. The frequency of KIR2DL1^+^/CD56^+^ and KIR2DL2/3^+^/CD56^+^ cells was 8.54–17.3% and 21.1–46.8%, respectively in four healthy volunteers (*n* = 8), indicating that the GiNKs demonstrated variable frequencies of KIR2DL1^+^ and KIR2DL2/3^+^ cells (Figure 1B).

### 2.3. HLA-C Expression in Gliomas in the Human Protein Atlas Data Set and GBM Cell Lines

To confirm the HLA-C expression patterns, we obtained the RNA sequencing data of gliomas from the Human Protein Atlas (HPA) database [38], which demonstrated that HLA-C was expressed in glioma tissues. The Kaplan-Meier curves following log-rank testing demonstrated that low HLA-C expression did not predict poor OS in the HPA database significantly (*p* = 0.20) (Figure 1C).

### 2.4. Effects of GiNKs Both Alone and in Combination with Antibodies (IgG and Anti-KIR2DLs) Treatment In Vitro

We investigated the growth inhibitory effects of the GiNKs both alone and in combination with IgG, anti-KIR2DL1 antibody (1–7F9), or anti-KIR2DL2/3 on U87MG and T98G cells using a real-time cell analysis (RTCA) system. 

GiNKs pre-incubated with IgG + IgG significantly inhibited U87MG and T98G cell growth from 3 h after treatment (effector-to-target; E:T = 0.5:1 and 1:1) (*p* < 0.001 in both cell lines). The growth inhibitory effects of GiNKs + IgG on the U87MG and T98G cells were clearly observed in a cell number-dependent manner (*p* < 0.001 in both cell lines) (Figure 2A). GiNKs pre-incubated with anti-KIR2DL1 antibody + anti-KIR2DL1 antibody also significantly inhibited U87MG and T98G cell growth from 3 h after treatment as compared to the vehicle alone. However, GiNKs + anti-KIR2DL1 antibody demonstrated significantly weaker inhibitory effects on U87MG and T98G cell growth as compared to control group, GiNKs pre-incubated with IgG + IgG from 24 h and 48 h after treatment, respectively (Figure 2B). Compared to the vehicle alone, GiNKs pre-incubated with control IgG alone significantly inhibited U87MG and T98G cell growth after 3 h from treatment (*p* < 0.001 in both cell lines). Similarly, GiNKs pre-incubated with anti-KIR2DL1 antibody alone significantly inhibited U87MG and T98G cell growth after 3 h from treatment (*p* < 0.001 in both cell lines) as compared to the vehicle alone. However, GiNKs pre-incubated with anti-KIR2DL1 antibody alone exhibited significantly weaker inhibitory effects on U87MG cell growth in comparison of GiNKs pre-incubated with IgG alone after 2 h from treatment (*p* < 0.001), while did not on T98G cell growth (Figure 2C). We also assessed the growth inhibitory effects of IgG alone or anti-KIR2DL1 alone on U87MG and T98G cells. Although IgG alone or anti-KIR2DL1 alone do not significantly inhibit U87MG cell growth compared to the vehicle alone, they significantly inhibited growth of T98G cells from 24 h after treatment (*p* < 0.001 in both antibodies) (Figure 2D). GiNKs pre-incubated with anti-KIR2DL2/3 antibody + anti-KIR2DL2/3 antibody(upper), only GiNKs pre-incubated with anti-KIR2DL2/3 antibody (middle), and only anti-KIR2DL2/3 antibody (lower) did not exhibit T98G cell growth in anytime compared to the GiNKs pre-incubated with IgG + IgG (upper), only GiNKs pre-incubated with IgG (middle), and only IgG (lower), respectively (Figure 2E). The different responses of U87MG and T98G cells to these antibodies suggest that these two cell lines are different character. The apoptosis detection assays demonstrated that GiNKs + IgG and GiNKs + anti-KIR2DL1 antibody induced U87MG and T98G cell apoptosis at 24 h. GiNKs + IgG significantly enhanced apoptosis induction of U87MG and T98G cells as compared to the vehicle alone. Additionally, GiNKs + IgG induced significantly increased the apoptotic cell population of U87MG and T98G cells at 24 h as compared to GiNKs + anti-KIR2DL1 antibody (Figure 3).

### 2.5. Effects of GiNK Treatment on an Orthotopic GBM-like Cell-Derived Xenograft Model

We evaluated the antitumor effects of GiNKs against intracranial orthotopic xenografts derived from an U87MG model using NOG mice in vivo. U87MG cells were implanted into NOG mouse brains, followed by intracranial infusions (treatments) via the same burr hole of the U87MG implantation (Figure 4A). The IgG group was significantly associated with a longer survival time compared to the NB group (*p* = 0.037) (Figure 4B). However, the anti-KIR2DL1 group was not significantly associated with survival time compared to the other groups (vs. NB group; *p* = 0.70, vs. IgG group; *p* = 0.16).

### 2.6. Histological Analysis

The histological analysis revealed that the tumors from all groups exhibited human GBM-like histological features at the time of autopsy (Figure 3). 

## 3. Discussion

As GBM has proven highly resistant to standard treatments [10], immunotherapy could be a novel and effective treatment thereof. Although the success of ICIs in malignant tumors such as melanoma and non–small cell lung cancer rapidly increased interest in immunotherapy as an alternative treatment for GBM, several phase 3 clinical trials of ICI drugs against newly diagnosed and recurrent GBM reported that they were not effective [6,7,8,9]. One possible reason for the inefficacious treatment is that GBM is a more aggressive, strongly heterogeneous, immunologically “cold”, and rapid-progression tumor [8,9]. One factor believed to contribute to the failure of ICI therapy is the primary focus of the treatment on single targets, such as chimeric antigen receptor-modified T cell–epidermal growth factor receptor variant III (CART-EGFRvIII) cells. Nevertheless, new approaches to resolve the issue of cancer immune evasion have been reported. A novel immunotherapy with vaccine therapy using oncolytic herpes virus G47Δ was effective for patients with residual or recurrent GBM after radiation therapy and temozolomide [39,40,41].

We have long focused on immunotherapy with NK cells as a novel therapeutic approach and have reported its results [15,17,42,43]. Immunotherapy utilizing NK cells can recognize GBM through mechanisms different from that in T cell-based therapies. Compared to T cell immunotherapy, NK cell immunotherapy of GBM has received less attention. NK cells recognize cancer cells without antigen sensitization [44] and can remove the abnormal cells as part of the innate immune system response [45,46]. NK cells exhibit potent cytotoxic activity against tumor cells by inducing apoptosis [14] and act as immune surveillance and suppress tumor development, proliferation, and metastasis [47,48]. The antitumor potential of NK cells has been suggested for more than 40 years. However, their clinical activity remains difficult to predict, responses are often not sustained, and their efficacy against tumors beyond acute myeloid leukemia is unclear [49,50]. The antitumor efficacy of NK cells has not been applied clinically. Previously, we expanded human peripheral blood NK cells harvested using a novel culture system for clinical application as GiNKs and reported a strong antitumor effect on GBM in vitro [15,16]. Our selective expansion method for autologous human NK cells is a simple, chemically defined, and feeder-free method that achieves the highest purity and greatest expansion scale. GiNKs have the potential to overcome the issues of previous NK cell-based immunotherapies. Furthermore, we have reported the antitumor effects of GiNKs on GBM cell lines in vitro. In the present study, we verified the antitumor effect of GiNKs on GBM cells via real-time assays. We confirmed that GiNKs exerted 46.9% and 57.2% growth inhibition against U87MG cells and T98G cells, respectively, at 3 h at an E:T ratio of 1:1. Our findings indicated that GiNKs exerted immediate cytotoxic efficiency on GBM cell lines in vitro. Moreover, we have reported an antitumor effect of GiNKs against subcutaneously implanted GBM-like ectopic xenograft models in vivo [17]. In the present study, we confirmed that direct intracranial infusion of NK cells in the form of GiNKs exerted antitumor effects on an orthotopic xenograft murine model in vivo. Altogether, these findings indicated that NK cell-based immunotherapy using GiNKs could represent a promising novel treatment option for GBM.

The balance of signals from the inhibitory and activating receptors on NK cells regulates their cytotoxic function against tumor cells [13,51]. The ligands for NK inhibitory receptors, such as PD-1, NKG2A, and KIR2DL, were associated with NK cell cytotoxicity against tumor cells [11,13,52]. Since KIR2DL1 and KIR2DL2/3 are both inhibitory receptors, we have considered blockade of these inhibitory receptors result in enhance antitumor effects of GiNKs in vitro and in vivo. Moreover, blockade of inhibitory KIR with IL-2 triggering reversed the functional hypoactivity of tumor-derived NK cells in GBM [53]. However, the present study demonstrated that GiNKs + KIR2DL1 blockade using 1–7F9 did not exhibit significantly enhanced antitumor effects but rather attenuated antitumor effects of GiNKs in vitro and in vivo. The extracellular domains of some activating KIRs demonstrate high sequence homology to those of some inhibitory KIRs (e.g., KIR2DS1–KIR2DL1, KIR2DS2–KIR2DL2, and KIR3DS1–KIR3DL1 pairs) [54]. Anti-KIR2DL1 antibody might also inhibit the effect of KIR2DS1 expressed on GiNKs. While the inhibitory KIR2DL1 binds to all HLA-C C2 with high avidity, the activating KIR2DS1, which is expressed on NK cell clones, also recognizes HLA-C C2 albeit with lower affinity than KIR2DL1 [55]. Consequently, anti-KIR2DL1 antibody might also inhibit the effect of KIR2DS1 expressed on GiNKs. The KIR2DL1 blockade did not enhance the antitumor effects; rather, it attenuated it in the same manner as in the in vitro experiment. Moreover, KIR2DL1 expression tends to be lower in GBM tissue than in non-tumor tissue, as demonstrated by the GlioVis data portal and TCGA database. Following log-rank testing, Kaplan-Meier curves revealed that low KIR2DL1 expression predicted significantly poor OS (*p* = 0.04), which might support the present findings. Furthermore, our group previously reported that PD-1-blocking antibodies did not exert an additive effect with GiNKs to prolong the survival of xenograft murine models bearing subcutaneous U87MG-derived tumors [17]. Therefore, our findings suggest that blocking antibodies against inhibitory ligands or receptors may not exert an additive effect with GiNKs against GBM cells. This supposition should be examined via scientific evaluation of many other inhibitory receptors expressed on NK cells. The direct intracranial infusion of GiNKs was highly effective against xenograft GBM cells independent of the KIR pathway. Nevertheless, the role of the KIR pathway in GBM remains controversial. It is possible that the GBM immunosuppression system against NK cells increases complexity through evasion via another pathway, for example, NKG2D engagement on human NK cells leads to DNAM-1 hypo-responsiveness [55]. Alternatively, KIR2DL1 might exert opposing effects on the inhibitory and activating systems in T cells [56]. Similarly in GiNKs, stimulation from KIR2DL1, which is typically an inhibitory system, contributes to NK cell activation, and inhibiting KIR2DL1 might have attenuated the anti-tumor effect of NK cells against GBM.

To our knowledge, this is the first report to evaluate the antitumor effect of GiNKs as highly purified human NK cells against an orthotopic xenograft murine GBM model. We previously reported antitumor effects in ectopic subcutaneous xenograft models derived from the U87MG GBM-like cell line [17]. In our orthotopic xenograft GBM model, NOG mice were intracranially implanted with U87MG cells, then GiNKs were subsequently intracranially infused via the same burr hole. Nonetheless, the anti-tumor efficacy of feeder cell-derived NK cells on GBM was recently documented [57]. Our GiNKs exhibited a potent anti-tumor effect compared to the NK cells expanded by a feeder cell-based culture. Furthermore, the GiNKs were readily accessible and of greater clinical utility compared to NK cells.

The blood–brain barrier (BBB) significantly reduces the efficacy of existing systemic therapies [58]. A novel immunotherapy with vaccine therapy using intratumoral administration of oncolytic herpes virus G47Δ was effective for patients with residual or recurrent supratentorial GBM after radiation therapy and temozolomide [39,40]. Direct intracranial injection of GiNKs also exerted specific antitumor effects, circumventing the BBB and increasing efficacy. The present study demonstrated that intracranial direct infusion of GiNKs prolonged the OS of the orthotopic xenograft GBM murine model. Moreover, the histological features at the time of autopsy supported the OS results. We simulated surgical removal of the tumor and continued local immunotherapy with intracranial injection of GiNKs and conducted the present research. Additionally, the implantation of fibrin gel as a T cell delivery system following tumor resection was reported recently and may enhance the efficacy of GiNKs and lead to practical treatment [59]. Finally, the present study demonstrated that the IgG group had significantly prolonged OS compared with the NB and anti-KIR2DL groups. Our findings suggested that intracranial direct infusion of GiNKs was highly effective against the orthotopic GBM-like cell-derived xenograft model. Moreover, various immunotherapy studies using allogeneic NK cells were reported recently [60,61]. NK cells do not require HLA matching. Autologous and allogeneic NK cells have the potential to overcome graft versus host disease. In addition, several clinical trials have exhibited the safety of allogeneic NK cell transfer [62]. Moreover, our GiNKs are allogeneic NK cells. To overcome the limitation of autologous NK cell-based immunotherapy, immunotherapy using allogeneic NK cells may be a treatment for patients with GBM.

Our study has some limitations. First, we used PBMCs derived from healthy volunteers. Typically, inducing GiNKs from the blood of patients with GBM is challenging due to the possibility of the patients having an immune function disorder [63]. Second, we used GBM cell lines, are known for their low HLA expression and not GBM patient tissue-derived cells, which do not reflect the heterogeneity of patient GBM. Finally, alkylating agents of cancer treatment such as temozolomide generally inhibit hematopoietic stem cell proliferation and limit lymphocyte numbers in the periphery [2]. In this situation, NK cells may also demonstrate decreased reactivity. It is necessary to investigate whether it is possible to induce GiNKs using blood from patients with GBM, as it is possible that the adoptively transferred GiNKs might exhibit limited persistence.

## 4. Materials and Methods

### 4.1. Reagents

Recombinant human anti-KIR2DL1 antibody (clone 1–7F9) was purchased from Creative Biolabs (Shirley, NY, USA). Anti-KIR2DL2/3, anti-human (clone DX27) was purchased from Miltenyi Biotech (Bergisch, Gladbach, Germany). The isotype control human IgG antibody (clone CB4) was purchased from Medical and Biological Laboratories (MBL Co. Ltd., Tokyo, Japan). 1–7F9 represents a humanized antibody with cross reactivity for KIR2DL1 and KIR2DL2/3. 

### 4.2. GBM Cell Lines

The human U87MG GBM-like cell line was from American Type Culture Collection (Manassas, VA, USA). The T98G GBM cell line was from RIKEN BioResource Research Center (Tsukuba, Japan). The cells were maintained in Dulbecco’s modified Eagle’s medium (Life Technologies, Carlsbad, CA, USA) supplemented with 10% heat-inactivated fetal bovine serum (MP Biomedicals, Tokyo, Japan), 100 U/mL penicillin, and 100 µg/mL streptomycin (Thermo Fisher Scientific, Waltham, MA, USA) at 37 °C in a humidified 5% CO_2_-containing atmosphere.

### 4.3. Animals

Six-to-eight-week-old female nonobese diabetes/severe combined immunodeficiency/gc null (NOD/SCID/NOG) mice were purchased from the Central Institute for Experimental Animals (Kanagawa, Japan). Mice were housed in specific pathogen-free environment under the condition of 12 h light/12-h dark cycle, free access to food and water. All procedures including care of mice, were performed in accordance with Policy on the Care and Use of Laboratory Animals, Nara Medical University and Animal Research: Reporting of In Vivo Experiments (ARRIVE) guidelines. The study was approved by The Animal Care and Use Committee in Nara Medical University (Number; 13403).

### 4.4. Expansion of GiNKs

The highly purified primary NK cells derived from human peripheral blood were expanded as described previously [15]. Briefly, peripheral blood mononuclear cells (PBMCs) were obtained from 24 mL heparinized peripheral blood from four healthy volunteers (mean age, 36.25 years). CD3-depleted PBMCs were isolated using a RosetteSep™ Human CD3 Depletion Cocktail (STEMCELL Technologies, Vancouver, Canada) according to the manufacturer’s prescribed protocol. Initially, 24 mL heparinized peripheral blood was mixed with 120 µL (50 µL/mL) of RosetteSep™ Human CD3 Depletion Cocktail, followed by incubation for 20 min at room temperature (25 °C). Next, 24 mL heparinized peripheral blood was diluted with 24 mL (equal to sample volume) of AIM V medium (Life Technologies, New York, NY, USA) and mixed gently. Then, the diluted sample was carefully layered on top of the density gradient medium by adding it to a tube containing 24 mL (equal to primary sample volume) of Lymphoprep™ (STEMCELL Technologies), being careful to minimize their mixing. Further, the 72 mL of sample was centrifuged at 1200× *g* for 20 min. Subsequently, the resulting enriched cell layer (PBMCs) was harvested and transferred to new tubes, and these PBMCs were mixed with AIM-V medium (Life Technologies) and centrifuged at 1700× *g* for 7 min. Finally, the isolated cells (approximately 1.0–3.0 × 10^7^ cells) harvested were placed in a T25 culture flask (Corning, Steuben, NY, USA) containing 10mL of AIM V medium (Life Technologies, New York, NY, USA) supplemented with 10% autologous plasma, 50 ng/mL recombinant human interleukin (rhIL)-18 (Medical and Biological Laboratories Co., Ltd.; MBL, Nagoya, Japan), and 3000 IU/mL rhIL-2 (Primmune Inc., Kobe, Japan) at 37 °C in a humidified 5% CO_2_-containing atmosphere. The AIM V medium supplemented with 3000 IU/mL rhIL-2 was replenished as necessary until used. All GiNKs were incubated for from 7 to 10 days prior to use.

### 4.5. Determination of Surface Antigen Expression

The cells were stained with the appropriate antibodies and fixed in 1% paraformaldehyde containing phosphate-buffered saline (PBS) at 4 °C for >30 min. Data were obtained using a BD FACSCalibur flow cytometer (BD Biosciences, San Jose, CA, USA) and analyzed using FlowJo version 10 (BD Biosciences). The following antibodies were used: Alexa 488-labeled anti-CD56 (clone B159, BD Pharmingen, Franklin Lakes, NJ, USA), allophycocyanin (APC)-labeled anti-KIR2DL1 (REA284, Miltenyi Biotech), and APC-labeled anti-KIR2DL2/3 (DX27, Miltenyi Biotech, Bergisch, Gladbach, Germany). The isotype control was APC-labeled IgG (REA293, Miltenyi Biotech).

### 4.6. Growth Inhibition Assays

The growth inhibitory effect on the U87MG and T98G cells was investigated using xCELLigence real-time cell analysis (RTCA) S16 and DP instruments (ACEA Biosciences, San Diego, CA, USA). The procedure has been described previously [42]. Briefly, complete medium (100 μL) was added to each well on E Plate 16 (ACEA Biosciences) and background impedance was measured at 37 °C in a humidified 5% CO_2_-containing atmosphere. T98G or U87MG cells (2 × 10^4^) suspended in 50 µL complete medium were added to each well as the target cells and impedance measurement was recorded for 72 h. After 24 h, the GiNKs (50 μL) were added to each well as the effector cells in the defined effector-to-target (E:T) cell ratios. The GiNKs were pre-incubated with control IgG, anti-KIR2DL1, or KIR2DL2/3 antibody prior for 30 min to use. Then, the growth inhibition assays were performed in the presence of 1 µg/mL IgG, anti-KIR2DL1, or anti-KIR2DL2/3. KIR2DL2/3 bind to HLA-Cw1, -3, -7, and -8 [34]. Furthermore, the U87MG GBM-like cell line and the T98G human GBM cell line were positive for HLA-Cw5 and -Cw4/Cw7, respectively [35]. To detect growth inhibitory effect of KIR2DL2/3, the T98G human GBM cell line was only used as the target cell. 

### 4.7. Apoptosis Detection Assays

The apoptosis detection assays were performed using APC-conjugated annexin V and propidium iodide (PI) solution (both, BioLegend, San Diego, CA, USA) according to the manufacturer’s instructions. Briefly, the GiNKs were incubated with control IgG, anti-KIR2DL1, or anti-KIR2DL2/3 antibody for 30 min prior to use. Then, the U87MG and T98G cells were exposed to the GiNKs at E:T ratios of 1:1 in the presence or absence of 1 µg/mL control IgG, anti-KIR2DL1, or anti-KIR2DL2/3 for 24 h. Following incubation, the floating cells and detached cells were collected and washed twice by cold Cell Staining Buffer (BioLegend) and resuspended in Annexin V Binding Buffer (BioLegend) at a concentration of 10^6^ cells/mL. Then, 5 μL APC-conjugated annexin V and 10 μL PI solution were added to the cell suspension of 100 µL Binding Buffer. The cells were gently mixed and incubated for 15 min at room temperature (25 °C) in the dark. Finally, 400 μL Binding Buffer was added and the apoptotic tumor cells were detected with a BD FACSCalibur flow cytometer (BD Biosciences). The data were analyzed using FlowJo version 10 (BD Biosciences).

### 4.8. In Vivo Orthotopic Xenograft Assays

The in vivo orthotopic xenograft assay was performed as described previously [43]. Briefly, the mice were anesthetized by inhalation of isoflurane mixed with air (induction, 2.5%; maintenance, 1.5%) and fixed on a stereotaxic instrument for mice (SR-6M-HT, Tokyo, Japan). The mice were stereotactically infused with 2 µL native Hank’s buffered salt solution (HBSS) containing 10^5^ U87MG cells into the right thalamus (2 mm lateral and 2 mm posterior from the bregma, and 3 mm dorsoventral from the outer border of the cranium) using an infusion syringe pump (Harvard Apparatus, Holliston, MA, USA) mounted with a Hamilton syringe (33S-gauge needle). The mice were randomly assigned to three intracranial infusion groups: negative background (NB, HBSS only), IgG [NK cells pre-incubated with IgG (10^6^ cells) + control IgG (1 µg)], and anti-KIR2DL1 [NK cells pre-incubated with anti-KIR2DL1 (10^6^ cells) + anti-KIR2DL1 (1 µg)]. The cells and reagents prepared using the aforementioned settings were directly infused intracranially using an infusion syringe pump via the same burr hole used to implant the U87MG cells earlier. The infusion speed was 1 µL/min for both the U87MG and NK cells.

### 4.9. Histochemical Analysis

The intracranial tumors were fixed in 10% neutral-buffered formalin and embedded in paraffin. Sections (5-µm thick) were placed on glass slides and stained with hematoxylin and eosin (HE). Photographs were captured using a BX-710 microscope unit (KEYENCE, Osaka, Japan) at ×40 and ×200 magnification.

### 4.10. Statistical Analysis

All results are reported as the mean ± standard deviation (SD) or the mean ± standard error of the mean (SEM). The statistical analyses were performed using Prism 9 (GraphPad Software Inc., San Diego, CA, USA). The log-rank test was performed for statistical analysis of survival time. The statistical significance of differences was determined using one- or two-way analysis of variance (ANOVA), followed by Tukey’s test for multiple comparisons. All reported *p*-values were 2-sided and considered statistically significant at *p* < 0.05, *p* < 0.01, *p* < 0.001, and *p* < 0.0001.

## 5. Conclusions

We demonstrated that immunotherapy using expanded and activated NK cells, i.e., GiNKs, directly administered into the brain may be a promising immunotherapeutic alternative in patients with GBM. These results could influence the standard treatment of patients with GBM and lead to novel treatment strategies.

## Figures and Tables

**Figure 1 ijms-24-14183-f001:**
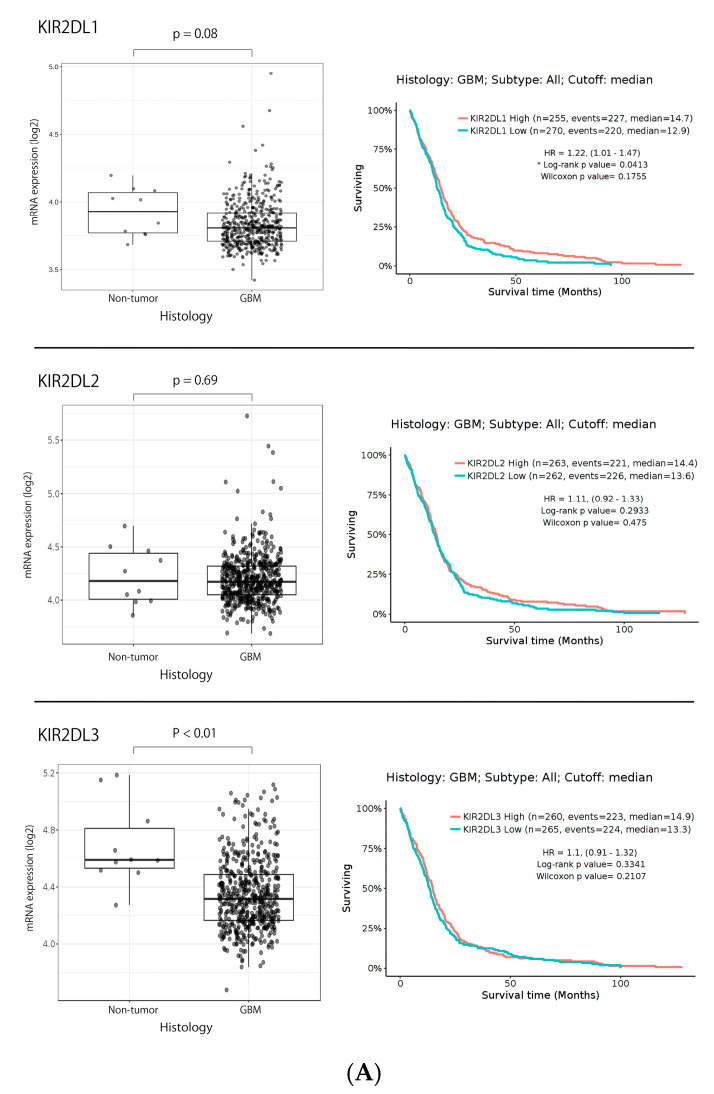
(**A**) KIR2DL1, KIRDL2, and KIR2DL3 expression patterns in GBM from the GlioVis data portal and TCGA database. **Left**: KIR2DL1 and KIR2DL2 were not significantly expressed in GBM tissues as compared to normal brain tissue in TCGA database (*p* = 0.08 and 0.69, respectively), while KIR2DL3 was significantly lower expressed in GBM (*p* < 0.01). **Right**: Kaplan-Meier curves based on mRNA expression from GlioVis data portal and TCGA database. The *p*-value was determined using Tukey’s honest significant difference test. Low KIR2DL1 expression predicted poor OS with significant differences in TCGA database (*p* = 0.04), while KIR2DL2 and KIR2DL3 expression did not (*p* = 0.29, and 0.33, respectively). * *p* < 0.05. (**B**) KIR2DL receptors expression pattern on GiNKs. **Left**: Representative flow cytometric data of KIR2DL1 and KIRDL2/3 expression on the GiNK surface. The top row was calculated based on the fluorescent intensity of APC-labeled IgG as an isotype control (Ctrl). The middle and bottom row were calculated based on the APC-labeled anti-KIR2DL1 and KIR2DL2/3, respectively. The KIR2DL1 and KIR2DL2/3 expression on the GiNKs were 11.5% and 32.4%, respectively. **Right**: The frequency of KIR2DL1^+^/CD56^+^ NK cells and KIR2DL2/3^+^/CD56^+^ NK cells were 8.54–17.3% and 21.1–46.8%, respectively, in four healthy volunteers. At least two independent experiments were performed (*n* = 8). (**C**) HLA-C expression pattern in glioma from the HPA data set. **Left**: Fragments per kilobase of transcript sequence per million base pairs sequenced (FKPM) value of HLA-C in gliomas. **Right**: Kaplan-Meier curves following log-rank testing demonstrating that low HLA-C expression did not predict poor OS in HPA database significantly (*p* = 0.20).

**Figure 2 ijms-24-14183-f002:**
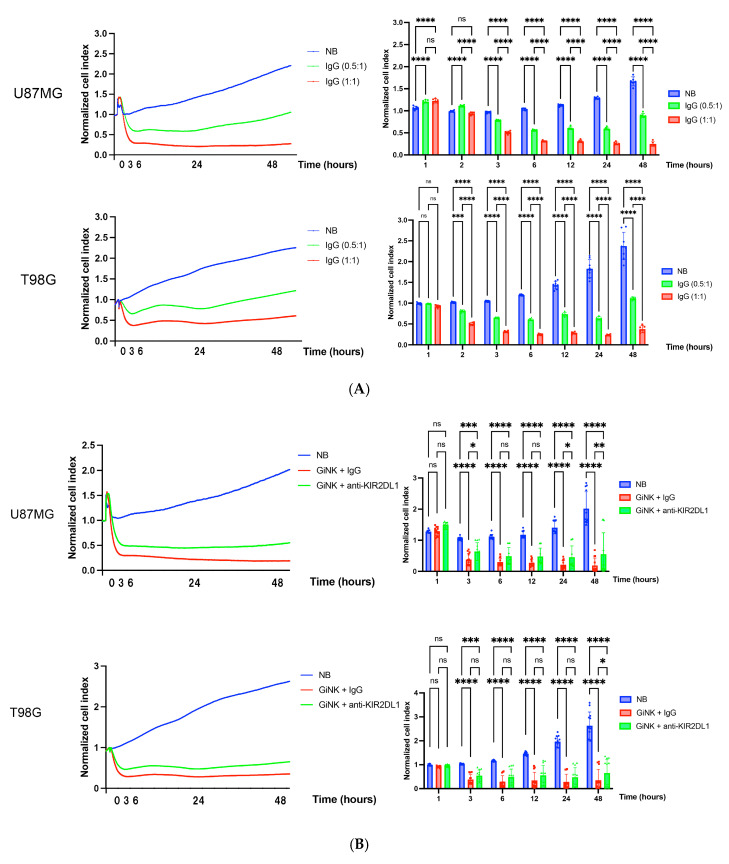
Enhanced growth inhibition of GBM cells by GiNKs. Images depict the real-time growth inhibition of U87MG (top) and T98G glioma cells (bottom) by GiNKs both pre-incubated with antibody, IgG, anti-KIR2DL1 (1–7F9) or anti-KIR2DL2/3, alone and in combination with antibody, and antibodies alone without GiNKs. The X- and Y-axes respectively depict the co-culture time and relative normalized cell index of each time point divided by the cell index of the co-culture start point. Data are the mean ± SEM. **Left**: Graphs illustrating representative data of real-time normalized cell index value of glioma cell lines. **Right**: Bar graphs illustrating the real-time cell analysis (RTCA)-based growth inhibition assays. Statistical differences were determined by two-way ANOVA followed by Tukey’s test. **** *p* < 0.0001, *** *p* < 0.001, ** *p* < 0.01, * *p* < 0.05, ns: not significant. (**A**) Representative data of real-time normalized cell index value of glioma cell lines co-cultured with GiNKs pre-incubated with IgG in combination with IgG at an effector-to-target (E:T) cell ratio of 1:1 (red) or 0.5:1 (green). The blue line indicates target cell lines only (NB). (*n* = 6, in two independent experiments in triplicate) (**B**) Representative data of real-time normalized cell index value of glioma cell lines co-cultured with GiNKs pre-incubated with IgG alone (red) or anti-KIR2DL1 antibody alone (green) at an E:T cell ratio of 1:1. The blue line indicates target cell lines only (negative background; NB) (*n* = 12, in four independent experiments in triplicate) (**C**) Growth inhibition of glioma cell lines by GiNKs pre-incubated with IgG or anti-KIR2DL1 antibody at an E:T cell ratio of 1:1 in combination with IgG (red) or anti-KIR2DL1 antibody (green), respectively. The growth curve (blue) indicates cell lines only (NB). (*n* = 12, in four independent experiments in triplicate) (**D**) Growth inhibition of glioma cell lines by IgG (red) or anti-KIR2DL1 (green) antibody alone, respectively. The growth curve (blue) indicates cell lines only (NB). (*n* = 6, in two independent experiments in triplicate) (**E**) GiNKs pre-incubated with anti-KIR2DL2/3 antibody + anti-KIR2DL2/3 antibody (green) (upper), only GiNKs pre-incubated with anti-KIR2DL2/3 antibody (green) (middle), and only anti-KIR2DL2/3 antibody (green) (lower) did not exhibit T98G cell growth in anytime compared to the GiNKs pre-incubated with IgG + IgG (red) (upper), only GiNKs pre-incubated with IgG (red) (middle), and only IgG (red) (lower), respectively. The blue line indicates target cell lines only (NB). (*n* = 9, 9, and 6 in three, three, and two, independent experiments in triplicate, respectively).

**Figure 3 ijms-24-14183-f003:**
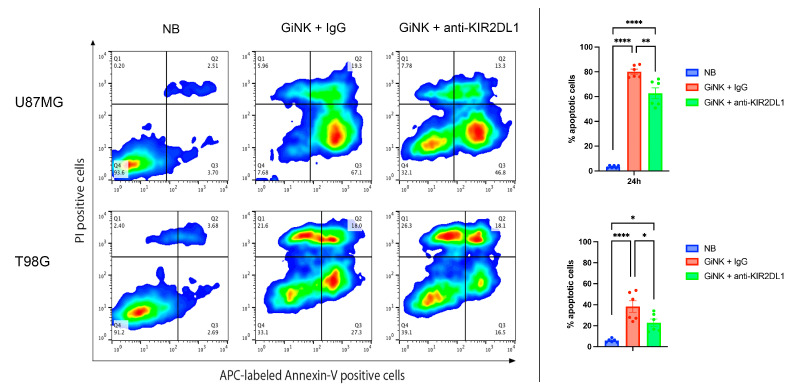
The percentage of apoptotic cells in U87MG (**top**) and T98G GBM cells (**bottom**) induced by target cell only; NB, GiNKs pre-incubated with IgG, and anti-KIR2DL1 antibody at 24 h. Left: Representative panels depicting the fluorescence intensity analysis of the PI and annexin V–APC fractions. Target cell only; NB (the **left** rows), GiNKs pre-incubated with IgG (the middle rows), and anti-KIR2DL1 antibody (the **right** rows). Right: Bar graphs illustrating the apoptotic cell analysis based on the flow cytometric data of APC-expressing cells. Statistical differences were determined by two-way ANOVA followed by Tukey’s test. **** *p* < 0.0001, ** *p* < 0.01, * *p* < 0.05.

**Figure 4 ijms-24-14183-f004:**
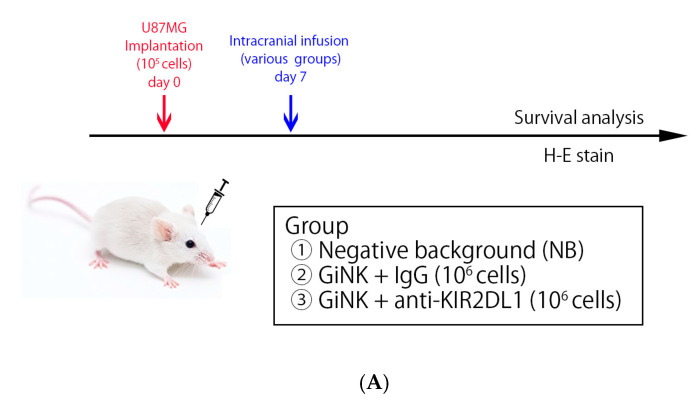
Effects of direct infusion of GiNKs pre-incubated with IgG or anti-KIR2DL1 in an orthotopic xenograft murine model with tumors derived from GBM-like cells. (**A**) Schematic of the experimental design. (**B**) Graph depicting the Kaplan–Meier curve. Blue, green, and red lines represent the NB group (*n* = 6), anti-KIR2DL1 group (*n* = 6), and IgG group (*n* = 6), respectively. The IgG mice were significantly associated with longer survival time compared to the NB mice (*p* = 0.037). The anti-KIR2DL1 mice were not significantly associated with survival time as compared to the other groups (vs. NB group; *p* = 0.70, vs. IgG group; *p* = 0.16). (**C**) Micrographs of HE staining overview (**top**) and ×400 (**bottom**) magnification. Left: Histocytological features at the time of autopsy of NB (**left**), anti-KIR2DL1 (**center**), and IgG (**right**) tumors. Scale bars, 50 µm. Statistical differences were determined by two-way ANOVA followed by Tukey’s test. * *p* < 0.05, ns: not significant.

## Data Availability

All relevant data is contained within the article: The original contributions presented in the study are included in the article, further inquiries can be directed to the corresponding authors.

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
