# Peer review of "Therapeutic Anti-KIR Antibody of 1–7F9 Attenuates the Antitumor Effects of Expanded and Activated Human Primary Natural Killer Cells on In Vitro Glioblastoma-like Cells and Orthotopic Tumors Derived Therefrom"

_ijms, 2023, doi:10.3390/ijms241814183_

Round 1

Reviewer 1 Report

The article submitted by Maeoka et al. it is well designed for most experiments and well argued. However, it raised the following concerns:

1. Two thirds of the experiments were conducted to verify the enhancement or not of the anti-tumor activity of GiNKs on GBM lines by antibodies against some KIRs, while nothing appears on this in the title. I understand that the authors wanted to emphasize the most important novelty of their results but they should strive to find a title more in keeping with the logic of their experiments/results.

2. Figure 2, panel E, shows results using only T98G cells and an anti-KIR2DL2/3 antibody, which does not appear among those listed in Materials and Methods. The authors should better explain which antibody it is and why the experiments were conducted only on T98G cells. Furthermore, it should be considered that with this antibody and those cells the authors obtained results similar to those observed in cells treated only with IgG.

3. In Figure 3, panels involving fluorescence images should carry an indication of the treatments to which the cells have been exposed, as in the graph to the right.

4. The discussion is quite long and sometimes repetitive. Above all, there is some confusion about the importance or otherwise of the treatment with antibodies against some KIRs. The authors should make the text on pages 12-13, lines 348-395, shorter and more incisive.

Minor points:

- at p. 5, line 168, the title of the paragraph should include the effects of GiNKs in the presence or absence of the antibodies used

- same page, line 171: the term RTCA is explained in the methods but in my opinion the acronym should be explained when it first appears in the text

Author Response

Thank you for the thoughtful and constructive feedback you provided regarding our manuscript, “Expanded and Activated Human Primary Natural Killer Cells Prolong the Overall Survival of Orthotopic Xenograft Models with Glioblastoma-like Cell-derived Tumors”.

Comments 1.

Two thirds of the experiments were conducted to verify the enhancement or not of the anti-tumor activity of GiNKs on GBM lines by antibodies against some KIRs, while nothing appears on this in the title. I understand that the authors wanted to emphasize the most important novelty of their results but they should strive to find a title more in keeping with the logic of their experiments/results.

Answer 1.

Thank you for this suggestion. We have revised the text that Therapeutic anti-KIR antibody of 1-7F9 attenuates the antitumor effects of Expanded and Activated Human Primary Natural Killer Cells on Orthotopic Glioblastoma-like Cell-derived Tumors in the title (P.1 lines 1-5).

Comment 2.

Figure 2, panel E, shows results using only T98G cells and an anti-KIR2DL2/3 antibody, which does not appear among those listed in Materials and Methods. The authors should better explain which antibody it is and why the experiments were conducted only on T98G cells. Furthermore, it should be considered that with this antibody and those cells the authors obtained results similar to those observed in cells treated only with IgG.

Answer 2.

Thank you for this suggestion. We have added the text that KIR2DL2/3 bind to HLA-Cw1, -3, -7, and -8 [34]. Furthermore, the U87MG GBM-like cell line and the T98G human GBM cell line were positive for HLA-Cw5 and -Cw4/Cw7, respectively [35]. To detect growth inhibitory effect of KIR2DL2/3, the T98G human GBM cell line was only used as the target cell. in the Materials and Methods (P.15 lines 505-508). We have also added the words, or anti-KIR2L2/3 in the Materials and Methods (P.15 lines 490, 491-2, 500, 502).

Comment 3.

In Figure 3, panels involving fluorescence images should carry an indication of the treatments to which the cells have been exposed, as in the graph to the right.

Answer 3.

Thank you for this suggestion. We have added the words that NB, GiNK + IgG, GiNK + anti-KIR2DL1 in the Figure 3. We have also added the texts that Target cell only; NB (the left rows), GiNKs pre-incubated with IgG (the middle rows), and anti-KIR2DL1 antibody (the right rows). in the Figure 3. legends (P.9 lines 256-257).

Comment 4.

The discussion is quite long and sometimes repetitive. Above all, there is some confusion about the importance or otherwise of the treatment with antibodies against some KIRs. The authors should make the text on pages 12-13, lines 348-395, shorter and more incisive.

Answer 4.

Thank you for this suggestion. We have revised and shortened the texts, and reduced the repetitions (P.12 lines 345-380).

Minor points:

Comment 1.

- at p. 5, line 168, the title of the paragraph should include the effects of GiNKs in the presence or absence of the antibodies used.

Answer 1.

Thank you for this suggestion. We have added the texts that both alone and in combination with antibodies (IgG and anti-KIR2DLs) in the title of the paragraph (P.5 lines 169-170).

Comment 2.

- same page, line 171: the term RTCA is explained in the methods but in my opinion the acronym should be explained when it first appears in the text

Answer 2.

Thank you for this opinion. We have revised and added the word that a real-time cell analysis (RTCA) in the Results (P.5 lines 173).

Reviewer 2 Report

In the work, „Expanded and Activated Human Primary Natural Killer Cells Prolong the Overall Survival of Orthotopic Xenograft Models with Glioblastoma-like Cell-derived Tumors“, the authors Ryosuke Maeoka et al. present a nice and informative work, aiming at a novel treatment strategy for glioblastoma multiforme.

The work is very interesting and innovative.

Some minor points may still be discussed or improved.

Of course, as the authors state themselves in the end of the discussion, long-term cultivated glioblastoma cell lines like U87MG and T98G do not represent real-life GBM, they are known for their low HLA expression. This should be explained a bit more in detail. Are there any data on fresh tumor-derived cells; what is their HLA expression pattern? And what is the HLA pattern in U87MG and T98G cells as compared to normal cells?

The HLA / KIR constellation may be of high significance, (eg concerning interaction between KIR2DL1and HLA Cw4).

What are the effects of the stimulated NK cells in healthy humans – is there possible a cytokine activation syndrome to be expected? The situation certainly is not comparable to the condition in immune deficient mice.

These limitations should be discussed.

How are the intracranial infusions exactly done; in which compartment do the cells go? How much damage is done to the brain?

Unfortunately, the language quality is not quite sufficient and the work should be worked over by a native speaker

see above

Author Response

Thank you for the thoughtful and constructive feedback you provided regarding our manuscript, “Expanded and Activated Human Primary Natural Killer Cells Prolong the Overall Survival of Orthotopic Xenograft Models with Glioblastoma-like Cell-derived Tumors”.

Comment 1.

Of course, as the authors state themselves in the end of the discussion, long-term cultivated glioblastoma cell lines like U87MG and T98G do not represent real-life GBM, they are known for their low HLA expression. This should be explained a bit more in detail. Are there any data on fresh tumor-derived cells; what is their HLA expression pattern? And what is the HLA pattern in U87MG and T98G cells as compared to normal cells? The HLA / KIR constellation may be of high significance, (eg concerning interaction between KIR2DL1 and HLA Cw4).

Answer 1.

Thank you for this opinion. We have added the texts that are known for their low HLA expression in the Discussion (P.13 lines 416). We have also added the text that KIR2DL2/3 bind to HLA-Cw1, -3, -7, and -8 [34]. Furthermore, the U87MG GBM-like cell line and the T98G human GBM cell line were positive for HLA-Cw5 and -Cw4/Cw7, respectively [35]. To detect growth inhibitory effect of KIR2DL2/3, the T98G human GBM cell line was only used as the target cell. in the Materials and Methods (P.15 lines 505-508). In addition, we are afraid to tell that we do not have any data on fresh tumor-derived cells from NOG mouse models in vivo.

Comment 2.

What are the effects of the stimulated NK cells in healthy humans – is there possible a cytokine activation syndrome to be expected? The situation certainly is not comparable to the condition in immune deficient mice. These limitations should be discussed.

Answer 2.

Thank you for this opinion and suggestion. However, our GiNKs are allogeneic NK cells as mentioned in the Discussion. In addition, we have added the citation and texts that NK cells do not require HLA matching. Autologous and allogeneic NK cells have the potential to overcome graft versus host disease. In addition, several clinical trials have exhibited the safety of allogeneic NK cell transfer [63]. in the Discussion (P.13 lines 407-409).

Comment 3.

How are the intracranial infusions exactly done; in which compartment do the cells go? How much damage is done to the brain?

Answer 3.

Thank you for this opinion. However, we have infused cells into the right thalamus of the brain stereotactically with a stereotaxic instrument for mice (SR-6M-HT, Tokyo, Japan) as mentioned in the Materials and Methods. In clinical practice, stereotactic surgery can infuse the compounds into the intended site 100% of the time. In addition, many studies have infused many cells or compounds into the brain in similar ways, so we have considered the damage to the brain is acceptable.

Comment 4.

Unfortunately, the language quality is not quite sufficient and the work should be worked over by a native speaker.

Answer 4.

Thank you for this suggestion. We have added the text that Dr. James Allen, from Oxford Science Editing Ltd.(https://www.oxfordscience.org) edited a draft of this manuscript in the Acknowledgments (P.16 lines 565-566). If you want to need the certification, we are going to submit the proof. Please don't hesitate to tell me if you need.

Again, thank you giving us the opportunity to strengthen our manuscript with your valuable comments and queries. We believe that we have addressed reviewers’ comments and hope that the revised manuscript is now acceptable publication in International Journal of Molecular Sciences. Thank you for your generous consideration.

Sincerely yours,

Round 2

Reviewer 1 Report

The authors mostly corrected the manuscript following the responses of the reviewers.

However, the title is not yet appropriate and should be of the type: "Therapeutic anti-KIR antibody of 1-7F9 attenuates the antitumor effects of expanded and activated human primary natural killer cells on in vitro glioblastoma-like cells and orthotopic tumors derived therefrom" (or similar)

They also forgot to indicate the source (commercial or not) from which they obtained the anti-KIR2DL2/3 antibody and I don't think they used the fluorescent antibody used for the determination of the surface antigen. Please correct.

Finally, the authors did not explain why the experiments with this antibody were conducted only on T98G cells and the importance of the fact that the effect obtained with this treatment in these cells is the same as that of cells exposed to GiNKs and IgG.

Author Response

Thank you for the thoughtful and constructive feedback you provided regarding our manuscript, “Expanded and Activated Human Primary Natural Killer Cells Prolong the Overall Survival of Orthotopic Xenograft Models with Glioblastoma-like Cell-derived Tumors”.

Comments 1.

The authors mostly corrected the manuscript following the responses of the reviewers.

However, the title is not yet appropriate and should be of the type: "Therapeutic anti-KIR antibody of 1-7F9 attenuates the antitumor effects of expanded and activated human primary natural killer cells on in vitro glioblastoma-like cells and orthotopic tumors derived therefrom" (or similar).

Answer 1.

Thank you for this suggestion. We have revised the text that Therapeutic anti-KIR antibody of 1-7F9 attenuates the antitumor effects of expanded and activated human primary natural killer cells on in vitro glioblastoma-like cells and orthotopic tumors derived therefrom in the title (P.1 lines 2-5).

Comment 2.

They also forgot to indicate the source (commercial or not) from which they obtained the anti-KIR2DL2/3 antibody and I don't think they used the fluorescent antibody used for the determination of the surface antigen. Please correct.

Answer 2.

Thank you for this suggestion. We have added the text that Anti-KIR2DL2/3, anti-human (clone DX27) was purchased from Miltenyi Biotech (Bergisch, Gladbach, Germany). in the Materials and Methods (P.13 lines 426-427).

However, we have performed flowcytometry analysis to detect KIR2DL2/3 expression on the GiNK cell surface (P.3 lines 122-125). So, we used the fluorescent antibody used for the determination of the surface antigen.

Comment 3.

Finally, the authors did not explain why the experiments with this antibody were conducted only on T98G cells and the importance of the fact that the effect obtained with this treatment in these cells is the same as that of cells exposed to GiNKs and IgG.

Answer 3.

Thank you for this suggestion. However, we have already mentioned that KIR2DL2/3 bind to HLA-Cw1, -3, -7, and -8 [34]. Furthermore, the U87MG GBM-like cell line and the T98G human GBM cell line were positive for HLA-Cw5 and -Cw4/Cw7, respectively [35]. To detect growth inhibitory effect of KIR2DL2/3, the T98G human GBM cell line was only used as the target cell. in the Materials and Methods (P.15 lines 494-497). We have considered these sentences are sufficient to answer your suggestion.

Again, thank you giving us the opportunity to strengthen our manuscript with your valuable comments and queries. We believe that we have addressed reviewers’ comments and hope that the revised manuscript is now acceptable publication in International Journal of Molecular Sciences. Thank you for your generous consideration.